# Drug survival and safety of biosimilars and originator adalimumab in the treatment of psoriasis: a multinational cohort study

Duc Binh Phan [1], Hugo Jourdain,[2] Alicia González-Quesada,[3] Mahmoud Zureik [2], Raquel Rivera-Díaz,[4] Antonio Sahuquillo-Torralba,[5] Miguel Angel Descalzo-Gallego,[6] Mark Lunt,[7] Ignacio Garcia-Doval,[6,8] Emilie Sbidian,[2,9] R B Warren,[1] Zenas Z N Yiu [1]

For numbered affiliations see end of article.

**Correspondence to**
Dr Zenas Z N Yiu;
zenas.yiu@manchester.ac.uk

## ABSTRACT

**Introduction** Psoriasis is a chronic inflammatory skin disease. Adalimumab is an effective but previously expensive biological treatment for psoriasis. The introduction of biosimilars following the patent expiry of the originator adalimumab Humira has reduced the unit cost of treatment. However, the long-term effectiveness and safety of adalimumab biosimilars for treating psoriasis in real-world settings are uncertain and may be a barrier to widespread usage.

**Methods and analysis** This study aims to compare the drug survival and safety of adalimumab biosimilars to adalimumab originator for the treatment of psoriasis. We will use both routinely collected healthcare databases and dedicated pharmacovigilance registries from the PsoNet initiative, including data from the UK, France and Spain. We will conduct a cohort study using a prevalent new user design. We will match patients on previous adalimumab exposure time to create two equal-sized cohorts of biosimilar and originator users. The coprimary outcomes are drug survival, defined by the time from cohort entry to discontinuation of the drug of interest; and risk of serious adverse events, defined by adverse events leading to hospitalisation or death. Cox proportional hazards models will be fitted to calculate HRs as the effect estimate for the outcomes.

**Ethics and dissemination** The participating registries agree with the Declaration of Helsinki and received approval from local ethics committees. The results of the study will be published in scientific journals and presented at international dermatology conferences by the end of 2023.

## STRENGTHS AND LIMITATIONS OF THIS STUDY

⇒ This is the largest real-world cohort study on adalimumab biosimilars' effectiveness and safety for psoriasis treatment.
⇒ The study incorporates healthcare data and pharmacovigilance registries from the UK, France, and Spain.
⇒ The study utilizes the prevalent new-user cohort design, including both switchers and new users, while accounting for previous adalimumab exposure to reduce selection bias.
⇒ Limitations of this observational study design include missing data, misclassification of biosimilars and originators, nocebo effect, and potential variations in biosimilar availability and data accuracy.

## INTRODUCTION

Psoriasis is a chronic inflammatory skin condition that affects approximately 2% of the global population.[1] Adalimumab is a commonly used systemic biological treatment for moderate to severe psoriasis.[2] With the patent for the originator adalimumab Humira expiring in Europe in October 2018, the introduction of biosimilars offered the potential to significantly reduce treatment costs. Despite being highly similar to the originator product, biosimilars are not identical due to their complex molecular structures and manufacturing processes.[3] However, the extrapolation of indications allows biosimilars to be approved for psoriasis based on extrapolated evidence of clinical equivalence in other diseases, even without being directly studied in clinical trials for psoriasis.[4]

In previous real-world comparative studies, no significant differences in drug retention, effectiveness and safety were found between biosimilars and Humira for inflammatory bowel diseases and rheumatic diseases.[5–8] However, studies investigating hidradenitis suppurativa treatment have suggested that switching from the originator to biosimilars was associated with increased risks of ineffectiveness and treatment discontinuation.[9 10] For the treatment of psoriasis, evidence is predominantly derived from short clinical trials, with only one small real-world study comparing biosimilars to Humira.[11]

This study shows that although there was no significant difference in treatment discontinuation, switching from Humira to GP2017 and SB5 was associated with more adverse events (AEs).[12] In general, the evidence regarding the usage of adalimumab biosimilars in real-world settings shows inconsistent results across different diseases and is limited to small retrospective cohort studies. The lack of robust real-world evidence raises concerns regarding long-term use, thus limiting the widespread adoption of adalimumab biosimilars for psoriasis.

Concerns about safety, efficacy, immunogenicity, extrapolated indication and lack of clinical data are the most commonly raised issues by clinicians in using biosimilars for psoriasis treatment.[13] Drug survival, defined as the length of time from initiation to discontinuation of therapy,[14] is a useful proxy for evaluating the long-term effectiveness and safety of adalimumab biosimilars in real-world settings. This evaluation can directly inform clinical decision-making and help identify the optimal treatment options for psoriasis. Furthermore, assessing the risk of AEs associated with adalimumab biosimilars in real-world settings is crucial to inform patients and healthcare providers, as clinical trials may not have adequately estimated this risk. The results of such evaluations can either reassure patients and clinicians of the effectiveness and safety of biosimilars and enhance confidence in their use or help identify any differences between currently available adalimumab products.

This study aims to compare the drug survival and safety of adalimumab biosimilars to adalimumab originator for the treatment of psoriasis in multiple national routine healthcare settings. We hypothesise that adalimumab biosimilars have no statistically and/or clinically significant difference in drug survival and safety compared with originator adalimumab on a population level.

## METHODS AND ANALYSIS
### Study design
This is a multinational federated cohort study using the prevalent new-user design illustrated by Suissa *et al*.[15] This protocol is reported following the HARmonized Protocol Template.[16] The study will be carried out from May 2023 and is expected to finish in December 2023.

### Patient and public involvement statement
An online survey was conducted to investigate the concerns, experiences and opinions of psoriasis patients on the use of biosimilars for psoriasis treatment. The survey was advertised through the UK Psoriasis Association research network, website and social media between 19 July and 19 August 2022. The results of the survey showed that 32 out of 36 (89%) respondents had significant concerns about the use of biosimilars for psoriasis treatment. Specifically, 25 (68%) respondents expressed concerns about the safety of these treatments, while 28 (78%) expressed concerns about their effectiveness. To further refine the research objectives, a focus group was consulted consisting of five patients who were already using biosimilar treatments for psoriasis.

The patient and public involvement activity yielded valuable insights into the significance of studying the long-term safety and effectiveness of biosimilars in real-world settings. These results were translated into research outcomes and were measured in our research where applicable. We will highlight the need for further research in our final report if the data available in the project was not sufficient to address any of these outcomes.

### Setting and variables
#### Data sources
This study comprises routinely collected healthcare data from the French Système National des Données de Santé (SNDS) and data from dedicated pharmacovigilance registries from the PsoNet network. PsoNet is composed of independent pharmacovigilance registries and healthcare databases focused on assessing the safety of biological therapy for patients with moderate-to-severe psoriasis. Two PsoNet studies provided data for this study: The British Association of Dermatologists Biologics and Immunomodulators Register (BADBIR) and the Spanish Registry of Systemic Therapy in Psoriasis (Biobadaderm). BADBIR and Biobadaderm have been previously described.[17 18] Briefly, BADBIR and Biobadaderm are longitudinal, multicentre, observational registers of patients with moderate-to-severe psoriasis who are receiving either conventional systemic or biological therapies for psoriasis. Despite variations in the design and monitoring, the registries have several common characteristics: encompassing all biological medications and all licensed systemic agents for psoriasis; monitoring patients for a predetermined period irrespective of the drug given; collecting details of demographics, concomitant comorbidities, current and previous systemic psoriasis treatments, changes in therapy, clinical assessments (AEs and disease severity) and self-reported outcome measures on registration and/or follow-up visit. The French SNDS covers almost the totality (>99%) of the French population—68 million residents. Each person is identified by a unique and anonymous number. The SNDS records comprehensive outpatient (procedures and pharmacy deliveries of reimbursed drugs) and inpatient (pharmacy deliveries of expensive drugs, procedures performed during hospital stays and discharge diagnoses coded according to the International Statistical Classification of Diseases and Related Health Problems, tenth revision, ICD-10) reimbursement information since 2006. The SNDS also contains sociodemographic information on sex, age, place of residence and vital status among others. Patients' status for 100% reimbursement of care related to a severe and costly long-term disease (LTD) is recorded and LTD diagnosis is coded according to the ICD-10. The SNDS has been extensively used to conduct pharmacoepidemiological studies, especially on the use, safety and effectiveness of health products.[19–21]

## Selection of study participants

Patients will be included accordingly to the following eligibility criteria:

► Patients of all ages and sexes.
► Patients with a diagnosis of plaque-type psoriasis (or psoriasis when the type of psoriasis cannot be specified).
► Patients on adalimumab originator Humira.
► Patients who initiated adalimumab treatment with adalimumab biosimilars: Amjevita/Amgevita/ Solymbic, Hyrimoz/Hefiya/Halimatoz, Idacio, Imraldi or switched from Humira to these biosimilars for non-medical reasons (defined as reasons other than efficacy, side effects or adherence). Patients with no classified reasons for switching will be considered as non-medical switch.
► Patients with at least one follow-up data entered after the first prescription of biosimilar/originator adalimumab for registries. For claims data, no minimal follow-up time will be required.
► Past and concomitant use of any topical therapies as well as conventional systemic treatments for psoriasis (including methotrexate, ciclosporin, acitretin, fumaric acid esters, hydroxycarbamide) is allowed.

Patients will be excluded following the study exclusion criteria:

► Patients with the presence of forms of psoriasis other than plaque-type (eg, pustular, erythrodermic or guttate psoriasis).
► Patients on biological treatments other than adalimumab.
► Patients with past and concomitant use of novel small molecule inhibitors including apremilast, deucravacitinib, tofacitinib.

## Interventions and comparators

The study interventions are initiating adalimumab treatment with biosimilars, including Amjevita/Amgevita/ Solymbic, Hyrimoz/Hefiya/Halimatoz, Idacio, Imraldi; or non-medical switching from adalimumab originator to adalimumab biosimilars.

The comparators are treatments with adalimumab originator Humira.

In registry databases, we define a starter as a patient who has not been previously exposed to adalimumab (known as adalimumab-naïve) and has started their first treatment with either originator or biosimilar adalimumab. A switcher is defined as a patient who has used adalimumab originator and then switched to adalimumab biosimilar, with at least one follow-up visit after the switch.

In the claims database, a starter is defined as a patient who starts adalimumab treatment after a washout period of 1 year without any previous treatment with adalimumab products. A switcher is defined as a patient who has received at least two consecutive deliveries of biosimilar adalimumab following originator treatment.

For patients with multiple adalimumab treatment sequences (ie, patients who switched from adalimumab

to other biologicals and later switched back to adalimumab) only the first adalimumab treatment sequence will be considered for inclusion in the study.

## Outcomes

The coprimary outcomes are discontinuation of treatment, defined by any gap in treatment for more than 90 days or changing to another biological treatment; and risk of serious adverse events (SAEs), defined by AEs leading to hospitalisation or death (table 1).

## Baseline and follow-up

At cohort entry, the following data will be collected where available: age, sex, presence of obesity, presence of psoriatic arthritis, comorbidities (excluding psoriatic arthritis, including any of hypertension, ischaemic heart disease, stroke, pulmonary fibrosis, asthma, chronic obstructive pulmonary disease, diabetes, thyroid disease, peptic ulcers, hepatic disease, renal disease, demyelinating disease, epilepsy, depression, tuberculosis, cancer), psoriasis onset time, concomitant topical treatments (start and stop date), concomitant non-biological systemic treatments (start and stop date), previous biological treatments (excluding recent Humira treatment for switchers), ongoing adalimumab treatments (dose regiments, start and stop date), disease severity at baseline (Psoriasis Area and Severity Index—PASI measured at the closest date to the cohort entry date, within 1 year before).

During follow-up records, the following data will be collected where available: concomitant non-biological systemic treatments (start and stop date), adalimumab treatments (dose regiments, start and stop date, reason for discontinuation), AEs (type of AEs and date recorded), SAEs (type of SAEs and date recorded). The study end date is 31 December 2022.

The maximum follow-up time will be determined according to the availability of data in each data source. We expect data for at least 1 year follow-up will be available in all involved databases. Registry databases follow-up participants during their healthcare visits or at least once a year. The SNDS follow-ups correspondingly to every reimbursed contact with health services.

## Data analysis

### Establishing analytical cohort

This study will implement the prevalent new-user cohort design with the following steps in delineating the analysis cohort. Separate pairs of biosimilar and matched originator cohorts will be identified for each biosimilar (ie, Amgevita vs Humira, Hyrimoz vs Humira, Imraldi vs Humira, etc).

1. Primary cohort formation: all eligible participants, including biosimilar and originator users, will be identified and included in the primary cohort.
2. Time 0 (T0) and exposure set identification: the first prescription of adalimumab for each user will be considered as T0 (day 0), and an exposure set will be formed for each month (each 30-day period) of adali-

**Table 1** Operational definitions of outcome

| Outcomes | Definitions | Estimations | Time frame |
|---|---|---|---|
| **Primary outcomes** | | | |
| Drug discontinuation—all causes | Discontinuation of biological therapy is defined as any gap in treatment for more than 90 days or changing to another biological treatment | ▶ Hazard function of discontinuation<br>▶ HR of discontinuation | 1, 2 or 3 years |
| Serious adverse event (SAE) | Serious adverse event is defined as an untoward medical occurrence that resulted in death, or hospitalisation (ie, at least one night of hospitalisation recorded). Adverse events are linked to a drug if they took place while the patient was using the drug. Details of the SAEs will be classified using the Medical Dictionary for Regulatory Activities (MedDRA) system | ▶ Incidence rate of developing SAE<br>▶ Incidence rate ratio of SAE<br>▶ Hazard function of developing the first SAE<br>▶ HR of the first SAE | From cohort entry up to the last available follow-up |
| **Secondary outcomes (these outcomes may not be available in all included data based and will only be measured where applicable)** | | | |
| Drug discontinuation due to treatment failure (where applicable—ie, for registry data) | Discontinuation of treatment that was documented as due to treatment failure. Patients with no specified reason for discontinuation are assumed as discontinued due to treatment failure | ▶ Cause-specific hazard function of discontinuation<br>▶ Cause-specific HR of discontinuation | 1, 2 or 3 years |
| Drug discontinuation due to adverse events (where applicable—ie, for registry data) | Discontinuation of treatment that was documented as due to adverse events. Patients with no specified reason for discontinuation, and records of SAE are assumed as discontinued due to adverse event | ▶ Cause-specific hazard function of discontinuation<br>▶ Cause-specific HR of discontinuation | 1, 2 or 3 years |
| Adverse event (AE) (where applicable—ie, for registry data) | An adverse event is defined as any untoward medical occurrence recorded while the patient was using the drug. Details of the AEs will be classified using the MedDRA system | ▶ Incidence rate of developing AE<br>▶ Incidence rate ratio of AE | From cohort entry up to the last available follow-up |

mumab treatment since T0. All biosimilar users who received their first biosimilar prescription at time T (T months since T0) and all originator users who continued to use the originator at time T will be included in the exposure set at time T.

3. Propensity score calculation: propensity scores will be calculated for each user to estimate the likelihood of using a biosimilar versus the originator adalimumab, using Cox proportional hazards model with time since T0 as the time variable, and age, sex, and calendar year of the first prescription as predictors.

4. Matching: within each exposure set, a 1:1 greedy matching algorithm will be used to match biosimilar users with originator users based on their propensity score, starting with the first chronological index biosimilar users and repeating for later users. Patients who have been selected as a matched comparator will not be included in subsequent matching as comparators (figure 1).

5. Study index date: the study index date will be the prescription date of the biosimilar and the corresponding prescription date of the biological in the exposure set.

6. Handling of patients who switch from originator to biosimilar: for originator users who are selected as comparators and later switch to a biosimilar, their follow-up will be censored at the time of switch, and they will be included as a biosimilar user from that point onwards. A matched comparator will be identified at the time of switch.

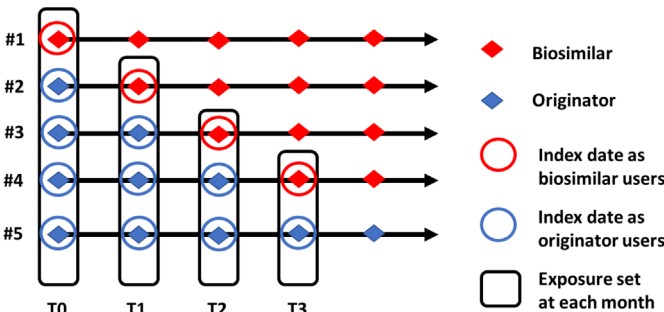

**Figure 1** Prevalent new-user cohort design. Example of the prevalent new-user cohort design with five participants, which are allocated to the exposures set at each month from T0 (0 month) to T3 (the third month).

## Statistical analysis

The drug discontinuation rate of adalimumab biosimilars and originator will be estimated for 1-year, 2-year or 3-year using Kaplan-Meier survival analysis. Censorship will occur at the last available follow-up date, at the end of follow-up, at the time originator users switched to adalimumab biosimilars, or at the time biosimilar users switched to another biosimilar. HR will be used as a measure of relative treatment effect. Cox proportional hazard models will be used to generate the HRs of the main exposures on drug discontinuation while adjusting for potential confounders. The proportional hazard assumption will be tested formally using Schoenfeld residuals. If the proportional hazard assumption is violated, we will consider alternative methods, including splitting the time by predefined follow-up time intervals (1 year) and considering using flexible parametric models. Where applicable (ie, for registry databases), reasons for discontinuation of each adalimumab biosimilar and originator will be analysed and classified as due to lack of effectiveness, due to AEs. Separate Cox proportional hazard models will be used to analyse overall discontinuation, discontinuation due to lack of effectiveness and discontinuation due to AEs.

SAEs and AEs will be assessed by calculating incidence rate per person years. The risk of SAEs and AEs will be compared by calculating the ratio (IRR) between the two comparator groups. The HRs for the first incidence of SAEs will be estimated using the Cox proportional hazard models, with censorship occurring at the last available follow-up date or the end of follow-up.

Multiple imputation models of 20 imputed datasets using chained equations will be performed to account for missing baseline data. This approach allows all participants to be included in the analysis, avoiding potential selection bias if only patients with completed data would be included.

## Multivariable regression model development

The final set of covariates to be included in the models will be determined based on data availability in each data source. An a priori list of covariates was predetermined. These are covariates that were associated with biological drug survival in psoriasis in previous studies and are expected to be available in all data sources, including age, sex, presence of obesity, psoriatic arthritis, number of comorbidities (excluding psoriatic arthritis), disease duration, length of adalimumab treatment, number of previous biological treatments (excluding recent Humira treatment for switchers) and concomitant non-biological systemic treatments (at baseline and during follow-up). In addition, calendar year of treatment initiation will be included in the models for adjustment.

For the analysis of SAEs, variables that will be included in the multivariable models are age, sex, presence of obesity, psoriatic arthritis, number of comorbidities (excluding psoriatic arthritis), disease duration, length of adalimumab treatment and concomitant non-biological systemic treatments.

## Meta-analysis

A meta-analysis of study results will be produced using a random-effects model. Each dataset will provide independent HRs and 95% CIs for the meta-analysis.[22] Estimated overall drug survival rates will be calculated using the pooled HRs and their corresponding±95% CI will be used to calculate best-case and worst-case drug survival rates. For the meta-analysis of IRR, the number of events and the person-years of exposure in each database will be pooled.

## Predefined subgroup analysis

► Each biosimilar will be analysed and compared with originator treatment separately.
► The incident new users (patients who started on biosimilars) and prevalent users (patients who switched from originators to biosimilars) will be analysed separately.
► Patients under the age of 18 and patients over 18 will be analysed separately.
► Registries data and routinely collected data will be grouped and analysed separately.

By conducting separate analyses for each biosimilar and patient subgroup, we can provide more precise and accurate estimates of biosimilar treatment among different patient populations and data sources.

## Sensitivity analysis

Sensitivity analyses that will be performed to add robustness to the results include, first, an analysis model in which patients with dose escalation will be regarded as treatment failure (discontinued due to inefficiency). The standard licensed dosing regimen for adalimumab is 40 mg every other week. Any increase in the prescribed dose or decrease in the dosing interval will be considered to be dose escalations. Second, a sensitivity analysis model will be conducted with censorships for all patients who discontinued their treatment with documented reasons of remission. Third, a model will be adjusted for baseline PASI, which will only be available from registry data. Forth, a sensitivity analysis stratifies switchers and corresponding continuous originator users as having less than 2 years and more than 2 years of adalimumab treatment before cohort entry. Fifth, a sensitivity analysis for AEs and SAEs uses a lag-time of 90 days, in which AEs are linked to a drug if they took place while the patient was using the drug or within 90 days after the end of exposure. In this model, patients who have been selected as a matched originator user will not be considered for the switcher cohort. Sixth, a sensitivity analysis uses inverse-probability-of-censoring adjusted regression to account for right-censored survival times with age, sex, length of adalimumab treatment and calendar year of cohort entry as predictors for censoring.

## Sample size considerations

Sample size is considered based on detecting or ruling out 0.5–1.5-fold differences in discontinuation rate as compared with the originator cohort (table 2).

## Data management and quality control

Data will be obtained and analysed independently for each database. No patient data will be exchanged between

**Table 2** Sample size considerations

| Type I error rate, α (two-tailed)=0.05; type II error rate, β=0.2; ratio of participants in two groups: 1:1. Assuming a baseline 1-year discontinuation rate of 25% in originator cohorts; a censoring rate of 30% per year in each cohort; and an average follow-up time of 3 years | Size of each cohort | Relative hazards can be detected/ruled out |
|---|---|---|
| | 112 | 0.5 |
| | 4000 | 0.9 |
| | 4546 | 1.1 |
| | 225 | 1.5 |

research teams. The final results, including number of participants, descriptive analysis of patients' characteristics and effect measures of outcomes will be combined in the meta-analysis.

## DISCUSSIONS

This study is anticipated to be the largest real-world cohort study investigating the effectiveness and safety of adalimumab biosimilars for the treatment of psoriasis. Using both routinely collected healthcare data and data from dedicated pharmacovigilance registries within the PsoNet initiative, this study draws on a diverse patient population from multiple countries, ensuring robust generalisability of study findings. To minimise selection bias and maximise the accuracy of study results, this investigation employs a prevalent new-user cohort design, which captures both switchers from adalimumab originator to biosimilars and patients who initiated treatment directly with adalimumab biosimilars. By accounting for previous adalimumab exposure, the study design controls for potential selection bias, confounders and enhances the validity of the observed outcomes. Overall, the findings of this study will provide valuable insight into the real-world effectiveness and safety of adalimumab biosimilars for psoriasis treatment, enabling informed clinical decision-making for patients and healthcare providers.

This study is subject to several limitations that may affect the validity of the results. First, the risk of bias due to missing data is a concern. The extent of bias depends on the type of missing data.[23] In the proposed study, we assume that baseline data are missing at random, that is, systematic differences between the missing values and the observed values can be explained by differences in observed data. Second, the accuracy of the data is dependent on the quality of the information documented in each database. Potential bias may arise due to misclassification of biosimilars and originator adalimumab. Therefore, it is critical to accurately document the drug brand name to ensure a valid comparison of biosimilars and originator in the study. Third, we assume that all switching from originator adalimumab to biosimilars are non-medical (switched for reasons other than efficacy, side effects or adherence). Violation of this assumption will lead to bias against the null hypothesis of the study.

Fourth, propensity score matching may reduce the sample size of the study due to the exclusion of subjects that could not be matched. An alternative approach to address this limitation is to use propensity score weighting when the sample sizes are limited. Fifth, in this study, patients and physicians were aware of the switch from the originator to biosimilars and initiation of biosimilars, potentially resulting in negative expectations or experiences with biosimilars. The lack of blinding is an inherent limitation in observational study designs that may introduce bias to the measured outcomes. Finally, the variations in the availability of biosimilars, in national policies of using biosimilars and variation in the design and data collection methods of databases would contribute to the heterogeneity in the outcomes measured across data sources. To investigate the potential heterogeneities, we will conduct predefined subgroup analyses stratified by patient groups and data sources.

## ETHICS AND DISSEMINATION
### Ethical considerations and protection of human subjects
The participating registries agree with the Declaration of Helsinki and received approval by local ethics committees: BADBIR: NHS Research Ethics Committee North West England, reference 07∕MRE08∕9; Biobadaderm: H 12 de Octubre. In France, EPI-PHARE has permanent regulatory access to the data via its constitutive bodies ANSM and CNAM, thus this present work did not require the approval from the French Data Protection Authority (CNIL).

### Plans for disseminating and communicating study results
The results of the study will be published in scientific journals by the end of 2023. The results will also be presented at international dermatology conferences. We will also present the study results to clinical communities, which involves presenting to patient communities.

**Author affiliations**
[1]Centre for Dermatology Research, Northern Care Alliance NHS Foundation Trust, The University of Manchester, Manchester Academic Health Science Centre, National Institute for Health and Care Research (NIHR) Manchester Biomedical Research Centre, Manchester, UK
[2]EPI-PHARE, French National Agency for Medicines and Health Products Safety (ANSM) and French National Health Insurance (CNAM), Saint-Denis, France
[3]Department of Dermatology, Hospital Universitario de Gran Canaria Dr Negrín, Las Palmas de Gran Canaria, Spain
[4]Department of Dermatology, Hospital Universitario 12 de Octubre, Madrid, Spain
[5]Department of Dermatology, Hospital Universitario y Politecnico La Fe, Valencia, Spain
[6], Research Unit, Fundacion Piel Sana AEDV, Madrid, Spain
[7]Versus Arthritis Epidemiology Unit, The University of Manchester, Manchester, UK
[8]Department of Dermatology, Complexo Hospitalario Universitario de Vigo, Vigo, Spain
[9]EpiDermE Epidemiology in Dermatology and Evaluation of Therapeutics, Assistance Publique-Hôpitaux de Paris (AP-HP), Paris Est Créteil University, Créteil, France

**Contributors** All authors substantially contributed to the design and writing of the protocol, gave final approval of the version to be published and agreed to be accountable for all aspects of the study. Conception and design: DBP, HJ, ES, MAD-G, IG-D, ML, RBW and ZZNY. Drafting of the article: DBP and ZZNY. Critical

revision of the article for important intellectual content: DBP, HJ, ES, AG-Q, MZ, RR-D, AS-T, MAD-G, IG-D, ML, RBW and ZZNY. Final approval of the article: DBP, HJ, ES, AG-Q, MZ, RR-D, AS-T, MAD-G, IG-D, ML, RBW and ZZNY. Supervision: ES, IG-D, RBW and ZZNY.

**Funding** The British Association of Dermatologists Biologic Interventions Register (BADBIR) is coordinated by the University of Manchester. BADBIR is funded by the British Association of Dermatologists (BAD). The BAD receives income from AbbVie, Almirall, Amgen, Celgene, Janssen, LEO Pharma, Lilly, Novartis, Samsung Bioepis, Sandoz Hexal AG and UCB Pharma for providing pharmacovigilance services. This income finances a separate contract between the BAD and the University of Manchester who coordinate BADBIR. All decisions concerning analysis, interpretation, and publication are made independently of any industrial contribution. In the UK, the research is funded by the Psoriasis Association and supported by the NIHR Manchester Biomedical Research Centre (NIHR203308). The BIOBADADERM project is promoted by the Fundación Piel Sana Academia Española de Dermatología y Venereología, which receives financial support from the Spanish Medicines and Health Products Agency (Agencia Española de Medicamentos y Productos Sanitarios) and from pharmaceutical companies (Abbott/Abbvie, Novartis, Lilly and Janssen). In France, the authors are employees of the French National Health Insurance (CNAM), the French National Agency for Medicines and Health Products Safety (ANSM) and the Assistance Publique—Hôpitaux de Paris (AP-HP) and received no funding for this study. The funding source did not intervene at any step of the study.

**Competing interests** RBW reported receiving research grants from AbbVie, Almirall, Amgen, Celgene, Janssen, Lilly, Leo, Novartis, Pfizer & UCB; and consulting fees from AbbVie, Almirall, Amgen, Arena, Astellas, Avillion, Biogen, Boehringer Ingelheim, Bristol Myers Squibb, Celgene, DiCE, GSK, Janssen, Lilly, Leo, Novartis, Pfizer, Sanofi, Sun Pharma, UCB & UNION. AG-Q acted as consultant and/or speaker for and/or participated in clinical trials for Abbvie, Pfizer, Novartis, Sanofi, Boeringher, Bristol-Meyer, Leo Pharma y Jansen. RR-D acted as consultant and/or speaker for and/or participated in clinical trials as IP for Abbvie, Almirall, Celgene, Janssen, Leo Pharma, Lilly, Novartis, MSD and Pfizer-Wyeth. AS-T has served as a consultant and/or paid speaker for and/or participated in clinical trials sponsored by companies that manufacture drugs used for the treatment of psoriasis, including AbbVie, Celgene, Janssen-Cilag, LEO Pharma, Lilly, Novartis and Pfizer. IG-D received travel grants for congresses from Abbvie, MSD and Pfizer.

**Patient and public involvement** Patients and/or the public were involved in the design, or conduct, or reporting, or dissemination plans of this research. Refer to the Methods section for further details.

**Patient consent for publication** Not applicable.

**Provenance and peer review** Not commissioned; externally peer reviewed.

**Open access** This is an open access article distributed in accordance with the Creative Commons Attribution 4.0 Unported (CC BY 4.0) license, which permits others to copy, redistribute, remix, transform and build upon this work for any purpose, provided the original work is properly cited, a link to the licence is given, and indication of whether changes were made. See: https://creativecommons.org/licenses/by/4.0/.

**ORCID iDs**
Duc Binh Phan http://orcid.org/0000-0002-8916-9093
Mahmoud Zureik http://orcid.org/0000-0002-8393-4217
Zenas Z N Yiu http://orcid.org/0000-0002-1831-074X

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
