## [Reviewer comments · BMJ Open]

ARTICLE DETAILS

TITLE (PROVISIONAL)	Protocol - Drug survival and safety of biosimilars and originator adalimumab in the treatment of Psoriasis: a multinational cohort study
AUTHORS	Phan, Duc Binh; Jourdain, Hugo; González-Quesada, Alicia; Zureik, Mahmoud; Rivera-Díaz, Raquel; Sahuquillo-Torralla, Antonio; Descalzo-Gallego, Miguel Angel; Lunt, Mark; Garcia-Doval, Ignacio; Sbidian, Emilie; Warren, R; Yiu, Zenas

VERSION 1 – REVIEW

REVIEWER	Kristensen, Salome Aalborg Universitetshospital, Department of Rheumatology
REVIEW RETURNED	08-Jun-2023

GENERAL COMMENTS	This is a protocol paper and the study aims to compare drug survival and safety of biosimilar Adalimumab with originator Adalimumab in patients with psoriasis. This is a very important clinical issue that needs attention and the study is a large real world cohort study and includes patients from different countries. I have som minor concerns. 1. Introduction: There are several studies for psoriatic arthritis and IBD comparing biosimilar with originator Adalimumab and bDMARD. Also few studies in Psoriasis. Please include their findings in the introduction.2. Page 8 - Baseline and follow-up section - have you considered including number of previous biological treatments? could be associated with drug survival3. Page 8 - Baseline and follow-up section - Is Napsi performed/collected. And what about arthritis disease activity with tender and swollen joints?4. Page 9 - statistical analysis - Please clarify how missing data are handled5. Design and discussion. Neither patients, nor physicians are blinded for the switch, and patient- and/or physician-related factors may affect outcomes. Thus, negative expectations towards the biosimilar (the so-called nocebo effect) and/or incorrect causal attributions, may potentially affect patients' experience and consequently outcomes. Previous studies have investigated, how the nocebo effect can be minimized by good informational and educational practices, Is nocebo effect considered in this study?
---

REVIEWER	Tobin, Anne Trinity College Dublin, Medicine
REVIEW RETURNED	11-Jun-2023

GENERAL COMMENTS	This is an interesting study, I suspect that the questions and analysis may change as the study progresses in line with recruitment, I believe the authors have done their utmost to identify this variability in methods and outcomes.
---

VERSION 1 – AUTHOR RESPONSE

Reviewers' Comments, Authors' Responses and Manuscript Changes

Reviewer #1

Comments to the Author:

This is a protocol paper and the study aims to compare drug survival and safety of biosimilar Adalimumab with originator Adalimumab in patients with psoriasis.

This is a very important clinical issue that needs attention and the study is a large real world cohort study and includes patients from different countries.

Response:

Thank you for your insightful comment. We appreciate your recognition of the importance of the study's objective, your appreciation and support are greatly valued.

Comment 1: *Introduction: There are several studies for psoriatic arthritis and IBD comparing biosimilar with originator Adalimumab and bDMARD. Also few studies in Psoriasis. Please include their findings in the introduction.*

Response:

We sincerely appreciate the reviewer's suggestion, and we fully agree to incorporate additional information from previous studies that compare adalimumab biosimilars with the originator.

Changes in the manuscript:

- Page 4 – Introduction: New paragraph added

“In previous real-world comparative studies, no significant differences in drug retention, effectiveness, and safety were found between biosimilars and Humira® for Inflammatory Bowel Diseases and Rheumatic diseases⁵⁻⁸. However, studies investigating Hidradenitis Suppurativa treatments have suggested that switching from the originator to biosimilars was associated with increased risks of ineffectiveness and treatment discontinuation^{9, 10}. For the treatment of psoriasis, evidence is predominantly derived from short clinical trials, with only one small real-world study comparing biosimilars to Humira®¹¹. This study shows that although there was no significant difference in treatment discontinuation, switching from Humira® to GP2017 and SB5 was associated with more adverse events¹². In general, the evidence regarding the utilisation of adalimumab biosimilars in real-world settings shows inconsistent results across different diseases and is limited to small retrospective cohort studies. The lack of robust real-world evidence raises concerns regarding long-term use, thus limiting the widespread adoption of adalimumab biosimilars for psoriasis.”

References: New references added:

5. Barberio B, Cingolani L, Canova C, et al. A propensity score-weighted comparison between adalimumab originator and its biosimilars, ABP501 and SB5, in inflammatory bowel disease: a multicenter Italian study. *Therap Adv Gastroenterol.* 2021;14:17562848211031420.

6. Casanova MJ, Nantes Ó, Varela P, et al. Real-world outcomes of switching from adalimumab originator to adalimumab biosimilar in patients with inflammatory bowel disease: The ADA-SWITCH study. *Aliment Pharmacol Ther.* 2023;58(1):60-70.
7. Larid G, Baudens G, Dandurand A, et al. Differential retention of adalimumab and etanercept biosimilars compared to originator treatments: Results of a retrospective French multicenter study. *Front Med (Lausanne).* 2022;9:989514.
8. Popescu CC, Mogoşan CD, Enache L, et al. Comparison of Efficacy and Safety of Original and Biosimilar Adalimumab in Active Rheumatoid Arthritis in a Real-World National Cohort. *Medicina (Kaunas).* 2022;58(12).
9. Grau-Pérez M, Rodríguez-Aguilar L, Roustan G, et al. Drug survival of adalimumab biosimilar vs adalimumab originator in hidradenitis suppurativa: Can equivalence be assumed? A retrospective cohort study. *Journal of the European Academy of Dermatology and Venereology.* 2023;37(5):e678-e80.
10. Burlando M, Fabbrocini G, Marasca C, et al. Adalimumab Originator vs. Biosimilar in Hidradenitis Suppurativa: A Multicentric Retrospective Study. *Biomedicines.* 2022;10(10).
11. Phan DB, Elyoussfi S, Stevenson M, et al. Biosimilars for the Treatment of Psoriasis: A Systematic Review of Clinical Trials and Observational Studies. *JAMA Dermatology.* 2023.
12. Loft N, Egeberg A, Rasmussen MK, et al. Outcomes Following a Mandatory Nonmedical Switch From Adalimumab Originator to Adalimumab Biosimilars in Patients With Psoriasis. *JAMA Dermatology.* 2021;157(6):676-83.

Comment 2: *Page 8 - Baseline and follow-up section - have you considered including number of previous biological treatments? could be associated with drug survival*

Response:

We agree with the reviewer's suggestion to include *the number of previous biological treatments* in our analytic models. We have revised the manuscript accordingly.

Changes in the manuscript:

- Page 8 – Baseline and follow-up

*At cohort entry, the following data will be collected where available: Age, sex, presence of obesity, presence of psoriatic arthritis, comorbidities (excluding psoriatic arthritis, including any of hypertension, ischaemic heart disease, stroke, pulmonary fibrosis, asthma, chronic obstructive pulmonary disease, diabetes, thyroid disease, peptic ulcers, hepatic disease, renal disease, demyelinating disease, epilepsy, depression, tuberculosis, cancer), psoriasis onset time, concomitant topical treatments (start and stop date), concomitant non-biologic systemic treatments (start and stop date), **previous biological treatments (excluding recent Humira® treatment for switchers)**, ongoing adalimumab treatments (dose regimens, start and stop date), disease severity at baseline (Psoriasis Area and Severity Index - PASI measured at the closest date to the cohort entry date, within 1 year before)*

- Page 9 – Multivariable regression model development

*The final set of covariates to be included in the models will be determined based on data availability in each data source. An a priori list of covariates was pre-determined. These are covariates that were associated with biologic drug survival in psoriasis in previous studies and are expected to be available in all data sources, including age, sex, presence of obesity, psoriatic arthritis, number of comorbidities (excluding psoriatic arthritis), disease duration, length of adalimumab treatment, **number of previous biologic treatments (excluding recent Humira® treatment for switchers)** and concomitant non-biologic systemic treatments (at baseline and during follow-up). In addition, calendar year of treatment initiation will be included in the models for adjustment.*

Comment 3: *Page 8 - Baseline and follow-up section - Is Napsi performed/collected. And what about arthritis disease activity with tender and swollen joints?*

Response:

We highly value the reviewer's suggestion regarding the inclusion of these variables in our models. However, it is important to note that our protocol was primarily designed for pharmacovigilance databases and claims databases focusing on psoriasis treatments. Unfortunately, the Nail Psoriasis Severity Index (NAPSI) data is not available in claims data, and arthritis disease activity is not typically captured in psoriasis registries. Therefore, regrettably, we would be unable to incorporate these specific variables into our study due to data limitations. We appreciate the reviewer's understanding of these limitations.

Comment 4: *Page 9 - statistical analysis - Please clarify how missing data are handled*

Response:

We agree with the reviewer's suggestion to provide additional details of how missing data will be handled in our study. We plan to employ multiple imputations to impute missing baseline data. This approach will not only allow us to handle missing data effectively but also help reduce the impact of selection bias.

Changes in the manuscript:

- Page 9 - Statistical analysis

Multiple imputation models of 20 imputed datasets using chained equations will be performed to account for missing baseline data. This approach allows all participants to be included in the analysis, avoiding potential selection bias if only patients with completed data would be included.

Comment 5: *Design and discussion. Neither patients, nor physicians are blinded for the switch, and patient- and/or physician-related factors may affect outcomes. Thus, negative expectations towards the biosimilar (the so-called nocebo effect) and/or incorrect causal attributions, may potentially affect patients' experience and consequently outcomes. Previous studies have investigated, how the nocebo effect can be minimized by good informational and educational practices, Is nocebo effect considered in this study?*

Response:

We appreciate the reviewer's insightful comment regarding the nocebo effect and for bringing this important aspect to our attention. As an observational study conducted in real-world settings, the absence of blinding is an inherent limitation in our study design. Nonetheless, we assure you that we will carefully consider the influence of the nocebo effect and any potential biases during the

interpretation and discussion of our study results. We have incorporated a section in our discussion that explicitly acknowledges the impact of the nocebo effect.

Changes in the manuscript:

- Page 9 - Statistical analysis

*This study is subject to several limitations that may affect the validity of the results. Firstly, the risk of bias due to missing data is a concern. The extent of bias depends on the type of missing data 23. In the proposed study, we assume that baseline data are missing at random, i.e. systematic differences between the missing values and the observed values can be explained by differences in observed data. Secondly, the accuracy of the data is dependent on the quality of the information documented in each database. Potential bias may arise due to misclassification of biosimilars and originator adalimumab. Therefore, it is critical to accurately document the drug brand name to ensure a valid comparison of biosimilars and originator in the study. Thirdly, we assume that all switching from originator adalimumab to biosimilars are non-medical (switched for reasons other than efficacy, side effects, or adherence). Violation of this assumption will lead to bias against the null-hypothesis of the study. Fourthly, propensity score matching may reduce the sample size of the study due to the exclusion of subjects that could not be matched. An alternative approach to address this limitation is to use propensity score weighting when the sample sizes are limited. **Fifthly, in this study, patients and physicians were aware of the switch from the originator to biosimilars and initiation of biosimilars, potentially resulting in negative expectations or experiences with biosimilars. The lack of blinding is an inherent limitation in observational study designs that may introduce bias to the measured outcomes.** Lastly, the variations in the availability of biosimilars, in national policies of utilising biosimilars and variation in the design and data collection methods of databases would contribute to the heterogeneity in the outcomes measured across data sources. To investigate the potential heterogeneities, we will conduct pre-defined subgroup analyses stratified by patient groups and data sources.*

Reviewer #2

Comments to the Author:

This is an interesting study, I suspect that the questions and analysis may change as the study progresses in line with recruitment, I believe the authors have done their utmost to identify this variability in methods and outcomes.

Response:

Thank you for your comment. We sincerely appreciate your recognition that the analysis may evolve as the study progresses and additional data becomes available.

VERSION 2 – REVIEW

REVIEWER	Kristensen, Salome Aalborg Universitetshospital, Department of Rheumatology
REVIEW RETURNED	30-Jun-2023

GENERAL COMMENTS

The response letter and the changes made in the manuscript is comprehensive.